# Interpretable integration of unpaired multi-omics for Alzheimer's diagnosis via cross-modal transformer reconstruction

Kai Liao[1,2,3,4☯], Danfeng Du[5☯], Jiawei Li[6], Jian Huang[7], Xiaodan Fan[1], Changshui Chen[2], Shanshan Wu[3], Bowei Yan[8*], Haibo Li[1,2,3*]

1 The Central Laboratory of Birth Defects Prevention and Control, The Affiliated Women and Children's Hospital of Ningbo University, Ningbo, China, 2 Ningbo Key Laboratory for the Prevention and Treatment of Embryogenic Diseases, The Affiliated Women and Children's Hospital of Ningbo University, Ningbo, China, 3 Ningbo Key Laboratory of Genomic Medicine and Birth Defects Prevention, The Affiliated Women and Children's Hospital of Ningbo University, Ningbo, China, 4 MOE Engineering Research Center of Gene Technology, School of Life Sciences, Fudan University, Shanghai, China, 5 Obstetrics and Gynecology Hospital of Fudan University, Shanghai, China, 6 School of Computer Science, Northwestern Polytechnical University, Shaanxi, China, 7 College of Computer Science, Chongqing University, Chongqing, China, 8 Institutes of Biomedical Sciences, Fudan University, Shanghai, China

☯ These authors contributed equally to this work.
* 22111510005@m.fudan.edu.cn, boweiyan2020@gmail.com (BY); lihaibo-775@163.com (HL)

## Abstract

Alzheimer's disease (AD) is a progressive neurodegenerative disorder with limited diagnostic tools and poorly understood molecular underpinnings. Although multi-omics technologies hold promise for early detection, integrating unpaired transcriptomic and epigenetic data remains a major challenge due to modality heterogeneity and small sample sizes. We present AE-Trans, an interpretable dual-channel Transformer framework that aligns RNA and DNA methylation data through cross-modal reconstruction and multi-head attention. AE-Trans achieves superior performance on prefrontal cortex datasets (accuracy = 0.9736, AUC = 0.9910) and demonstrates strong generalizability to external regions temporal cortex cohorts across brain regions (accuracy = 0.7389, AUC = 0.8432). To validate the performance within the same brain region, we tested AE-Trans on an external unpaired multi-omics dataset from the prefrontal cortex. Additionally, we validated the model on a paired multi-omics dataset to assess whether it could achieve good results in real-world scenarios. In the unpaired dataset from the external same brain region, AE-Trans achieved an accuracy of (accuracy = 0.87) and AUC of (AUC = 0.94), while in the real-world paired multi-omics dataset, the accuracy was (accuracy = 0.88) and AUC was (AUC = 0.93). These results demonstrate that AE-Trans not only validates well on external unpaired datasets, but also generalizes effectively to real-world multi-omics paired datasets, highlighting its robustness in practical applications. Through counterfactual integrated gradients, we identified key features associated with immune regulation, hormonal signaling, and neuronal metabolism. These were validated via

**Data availability statement:** Source data are provided with this paper. Source data is publicly available on Gene Expression Omnibus by accession numbers GSE33000, GSE44770 and GSE80970. Our postprocessed form of the these publicly available data is available at https://doi.org/10.5281/zenodo.13933763.

**Funding:** This work was supported by the Ningbo Municipal Bureau of Science and Technology (the General Project of Ningbo Public Welfare Research Program (2023S043),Innovation Project of Distinguished Medical Team in Ningbo (2022020405), Key Technology Breakthrough Program of 'Ningbo Sci-Tech Innovation YONGJIANG 2035'(2024Z222) and NINGBO Leading Medical &Health Discipline(No.2026-A34) to KL; the Social Development Public Welfare Foundation of Ningbo (2022S035), Ningbo Science and Technology Project (2023Z178), Major Research Project of Ningbo Clinical Medical Research Center (2024L002), Key Technology Breakthrough Program of 'Ningbo Sci-Tech Innovation YONGJIANG 2035' (2024Z221), and Ningbo Medical and Health Brand Discipline (PPXK2024-06) to HL; Key Technology Breakthrough Program of Ningbo Sci-Tech Innovation YONGJIANG 2035 (2025Z160) to SW) and the Zhejiang Provincial Health Commission (Clinical Innovation Team Talent Project(CXTD202502005) to KL). The other authors did not receive any financial support. The funders had no role in study design, data collection and analysis, decision to publish, or preparation of the manuscript. The authors received no specific funding for this work.

**Competing interests:** The authors have declared that no competing interests exist.

pathway enrichment and logistic regression (AUC = 0.9749), confirming the biological relevance of model-derived markers. Furthermore, AE-Trans generalized well to two independent RNA datasets, where latent representations not only improved classification (AUCs = 0.92 and 0.89) but also stratified patients into subgroups with significantly different prognoses. These results highlight AE-Trans as a robust and explainable tool for multi-omics integration, supporting early diagnosis, biomarker discovery, and individualized risk prediction in Alzheimer's disease.

## Author summary

With the increasing adoption of omics technologies in biomedical research, integrating multi-modal data from diverse sources has become crucial for understanding complex diseases and enabling precision diagnosis and treatment. We present AE-Trans, an interpretable dual-channel Transformer framework that integrates unpaired transcriptomic and epigenomic data to enhance Alzheimer's disease diagnosis. The model demonstrates robust performance across brain regions and datasets, achieving high accuracy on both paired and unpaired multi-omics data. Through explainable AI methods, AE-Trans identifies biologically relevant biomarkers associated with immune and metabolic pathways, while its learned representations enable patient stratification and prognostic prediction. This work offers a powerful tool for early diagnosis, biomarker discovery, and personalized risk assessment in neurodegenerative diseases.

## Introduction

The escalating global burden of Alzheimer's disease (AD)—a leading cause of death worldwide and a major contributor to disability in the elderly—underscores a pressing, unmet clinical need for early, molecular-level diagnostics [1–3]. Current approaches, primarily reliant on neuroimaging and cognitive testing, often identify AD only at advanced stages, limiting the window for effective intervention [1,2]. In contrast, transcriptomic and epigenomic profiling offer a promising avenue for uncovering early molecular signatures and enabling mechanism-driven diagnosis [4–6]. However, harnessing their full potential remains difficult due to the extreme high dimensionality, small sample sizes, and the pervasive lack of paired samples across modalities [7,8]. This unpaired nature creates a profound information chasm, preventing conventional models from capturing intricate inter-omic relationships essential for a holistic understanding of AD pathology.

Traditional machine learning (ML) models have shown utility in multi-omics integration, improving classification by combining genotype, transcriptome, and proteome data [9]. For example, Xia et al. [10] proposed a feature-fusion framework that revealed disease-associated brain regions (e.g., olfactory cortex) and genes (e.g., CNTNAP2, LRP1B). However, such methods often rely on handcrafted features,

struggle in high-dimensional low-sample-size (HDLSS) contexts, and provide limited biological interpretability—critical for clinical translation.

Recognizing these limitations, deep learning (DL) models have emerged as powerful alternatives. Sehwan et al. [11] integrated unpaired RNA and methylation data using deep matrix factorization (AUC = 0.801). Park et al. [12] combined DEG/DMP-driven feature selection with a DNN (AUC = 0.823). Zeeshan et al. [7]. Nivedhitha et al. [13] employed a deep belief network and Jaccard fusion (accuracy = 0.82). Qi et al. [14]. applied a hierarchical embedding strategy with knowledge distillation to highlight AD-sensitive regions like the hippocampus. Yet, existing DL models face common bottlenecks: (i) reliance on matched data, (ii) fragmented, non-end-to-end pipelines, (iii) poor interpretability, and (iv) limited ability to model long-range inter-modality dependencies—an area where Transformer-based architectures can excel.

To address these challenges, we introduce AE-Trans, a dual-channel Transformer framework designed to facilitate the integrative analysis of unpaired RNA and DNA methylation data in a robust and effective manner. AE-Trans combines modality-specific autoencoders with cross-modal Transformer encoders to capture complex patterns both within each modality and across different modalities in biological cohorts. A distinctive bidirectional reconstruction module aligns the latent representations across omics layers, enabling unprecedented feature fusion. To further enhance clinical relevance, we incorporate a counterfactual integrated gradients module for transparent attribution, facilitating biomarker discovery and mechanistic interpretation.AE-Trans consistently outperforms existing baselines. On the GSE33000 cohort, it achieved an AUC of 0.9910 and accuracy of 0.9736—surpassing AE-XGBoost by +0.047 in AUC. In this study, we demonstrate AE-Trans's remarkable ability to generalize across brain regions by testing it on three datasets: unpaired cross-region datasets, unpaired same-region datasets, and real paired datasets. This showcases the model's ability to effectively transfer knowledge from unpaired datasets to paired datasets across different brain regions and within the same brain region. First, we showcase the model's cross-brain-region generalization by testing it on external unpaired datasets from the superior temporal cortex (GSE132903 and GSE80970), where it achieved robust performance (AUC = 0.8432, accuracy = 0.7389), demonstrating AE-Trans's capability to handle data from regions different from the one it was trained on. To further validate its performance, we tested AE-Trans on an additional unpaired external dataset from the frontal cortex (GSE5281, GSE15222, GSE12685, GSE36980, GSE48350, GSE59685, GSE66351), where it again demonstrated strong generalization (AUC = 0.9396, accuracy = 0.8732). These results show that AE-Trans not only generalizes across regions but also maintains strong performance within the same brain region. Finally, AE-Trans was tested on a paired multi-omics dataset from the same frontal cortex region (GSE110732), where it achieved excellent results (AUC = 0.9267, accuracy = 0.88). This highlights AE-Trans's ability to successfully transfer knowledge from unpaired data to paired data, demonstrating its versatility and robustness in bridging the gap between these dataset types. To demonstrate the interpretability of AE-Trans, a logistic regression classifier built on the top 200 features identified by AE-Trans also achieved high predictive accuracy (AUC = 0.9749), underscoring the reliability of model-driven feature selection. Furthermore, the pretrained AE-Trans model fine-tuned on two independent RNA-seq datasets (GSE118553 and GSE29378) successfully distinguished AD from control samples and stratified patients into subgroups with distinct survival outcomes. These latent clusters also exhibited consistent transcriptional differences, as reflected in the expression heatmaps. These findings establish AE-Trans as a powerful and interpretable platform for early AD diagnosis and personalized prognosis, laying the foundation for precision therapeutics and targeted interventions.

## Materials and methods

### Data collection and preprocessing

We curated multi-omics datasets from six publicly available GEO studies: GSE33000 [15], GSE44770 [16]), GSE80970 [17], GSE132903 [18], GSE118553 [19], and GSE29378 [20,21], encompassing RNA-seq and DNA methylation profiles from the prefrontal cortex, middle and superior temporal gyri, hippocampus, and temporal cortex. To address potential batch effects across datasets collected from different sources and time points, RNA-seq data were batch-harmonized

using empirical Bayes methods (e.g., ComBat). RNA-seq data from GSE33000 and GSE44770, and methylation data from GSE80970, were used for model training and internal validation. For external evaluations, we used three distinct datasets to assess the model's performance and generalizability. First, we tested AE-Trans on a cross-region unpaired dataset, which included RNA data from GSE132903 and methylation data from the superior temporal gyrus in GSE80970, representing data from different brain regions. Second, to evaluate performance within the same brain region, we used an unpaired same-region dataset from the frontal cortex, consisting of 181 transcriptomes (GSE5281, GSE15222, GSE12685, GSE36980, GSE48350) and 147 methylomes (GSE59685, GSE66351). Finally, we assessed the model's ability to handle paired data by using a real paired dataset from the frontal cortex (GSE110732), which contained 25 matched transcriptome and methylation samples. Two additional RNA-seq datasets (GSE118553 and GSE29378) were used in a downstream fine-tuning task to evaluate the model's capacity for AD classification and phenotype stratification based on RNA features alone.

For RNA-seq preprocessing, samples from patients with Huntington's disease were excluded from GSE33000, yielding 467 samples (310 AD, 157 controls). GSE44770 contributed 230 samples (129 AD, 101 controls). Probe IDs were mapped to official gene symbols using platform-specific annotation files. Redundant probes (i.e., multiple probes mapping to the same gene) were averaged arithmetically. Probes without clear gene annotation were discarded. The resulting expression matrices from both datasets were aligned on 19,348 shared genes. Genes with >30% missing values were removed; remaining missing entries were zero-imputed to preserve sparsity. The final merged RNA training cohort included 697 samples (439 AD, 258 controls). GSE132903 was similarly processed, yielding 195 samples (97 AD, 98 controls) and 14,926 overlapping genes. GSE118553 (45 AD, 24 controls) and GSE29378 (31 AD, 32 controls) underwent identical harmonization.

For methylation preprocessing, we selected 142 prefrontal cortex samples (74 AD, 68 controls) from GSE80970 for training, and 144 samples (74 AD, 70 controls) from the middle temporal gyrus for external evaluation. CpG probes located within 1500 bp upstream of transcription start sites (TSS) were mapped to gene symbols. Duplicate mappings were averaged, and ambiguous probes removed. Features with >30% missing values were excluded. We then aggregated CpG-level data to gene-level methylation features by averaging all mapped probes for each gene, resulting in 2,039 high-confidence gene-level methylation features. To unify modalities, we extracted the intersection of gene symbols from RNA and methylation data, resulting in 14,926 common features.

Although these omics datasets are unpaired and derived from distinct brain regions, AE-Trans does not assume anatomical homogeneity. Instead, its architecture leverages multi-head attention and shared latent embeddings to model transferable biological signals across modalities and brain regions. Cross-region evaluation further demonstrated the model's robustness to anatomical heterogeneity.

## Data split and integration strategy

To integrate unpaired RNA and methylation datasets while preserving biological validity, we designed a combinatorial intra-label pairing scheme inspired by prior work on cross-modal feature correspondence) [12,22,23]. RNA and methylation samples were first split by label (AD vs. control) into four subsets: RNA-AD, RNA-control, Meth-AD, and Meth-control. Each subset was randomly divided into training and testing sets using an 80/20 split, strictly conducted at the sample level prior to any fusion, thus eliminating potential information leakage.

For training, RNA-AD samples were systematically matched with all Meth-AD samples to generate positive training pairs, and likewise for controls to form negative pairs. This exhaustive pairing yielded a balanced and biologically consistent training corpus. Test samples were paired using the same strategy. To mitigate redundancy and control computational complexity, we adopted mini-batch sub-sampling during training. Specifically, each mini-batch was generated by randomly sampling a subset of RNA samples and pairing each with a fixed number of randomly selected methylation samples from the same label group, thus maintaining label consistency while limiting batch size.

Although this pairing strategy introduces synthetic combinations, the shared diagnostic label and omics-specific encoders allow the model to focus on learning representative inter-modality associations. AE-Trans's attention mechanism and bidirectional reconstruction loss are designed to filter spurious co-occurrence patterns and reinforce biologically meaningful alignment in the latent space.This strategy allowed AE-Trans to effectively model relationships across independently sampled modalities, simulating real-world clinical settings where omics data are collected separately.

## Transformer-based omics fusion method (AE-Trans)

AE-Trans is a multimodal deep learning framework based on the Transformer architecture, tailored for integrative analysis of unpaired RNA and DNA methylation data in Alzheimer's disease research. As depicted in Fig 1, the pipeline consists of three major stages: (1) omics-specific encoding and dimensionality reduction, (2) latent feature alignment and fusion, and (3) downstream classification and interpretability. The architecture is composed of five functional modules: omics-specific encoding layers, abstraction via autoencoder-Transformer encoders, a classification head, a dual-path reconstruction unit, and a counterfactual attribution module.While referred to as a "multimodal integration module" in earlier drafts, this

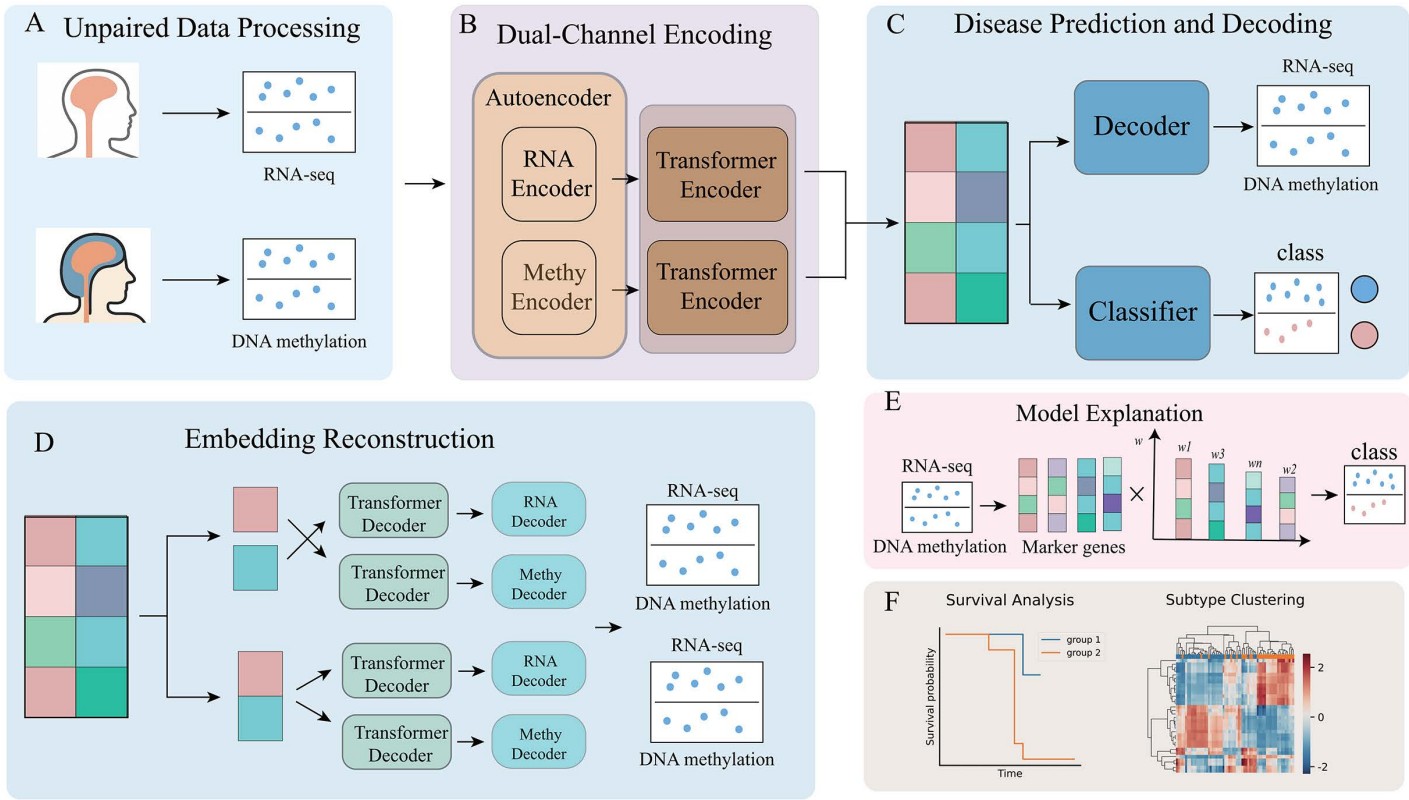

**Fig 1. AE-Trans model framework.** The architecture comprises modality-specific processing, cross-modal alignment, and downstream interpretability modules. (A) Data pairing strategy. Unpaired RNA and DNA methylation samples are integrated through a combinatorial intra-label pairing scheme based on diagnostic status (AD vs. Control). (B) Dimensionality reduction and encoding. High-dimensional inputs are compressed via modality-specific autoencoders, followed by Transformer encoders with multi-head attention to capture global dependencies. (C) Fusion and classification. Latent embeddings from dual channels are concatenated and passed through a linear fusion layer to a multilayer perceptron (MLP) for AD probability prediction. (D) Dual-path reconstruction. A bidirectional reconstruction unit aligns cross-modal representations using cycle-consistency loss, ensuring the preservation of biological signals across transcriptomic and epigenomic spaces. (E) Counterfactual attribution. Model interpretability is achieved via Counterfactual Integrated Gradients (CIG), quantifying the contribution of individual features to identify AD-related biomarkers. (F) Clinical validation. Latent representations enable patient stratification into subgroups with distinct survival outcomes and transcriptomic profiles, demonstrating the model's prognostic utility.

component does not represent a separate network layer. Instead, it encompasses the overall data preparation strategy: modality-specific encoders followed by pairwise sample construction across matched diagnostic labels (Fig 1A). These combinatorial pairings ensure statistical independence between training and test sets while modeling biological heterogeneity.

Each modality was processed by a separate autoencoder for dimensionality reduction, mitigating high-dimensionality bottlenecks. The resulting low-dimensional embeddings were passed into two independent multi-layer Transformer encoders (three layers each, with four attention heads per layer) to capture intra- and inter-modality dependencies (Fig 1B). Although the encoders operate separately, cross-modal dependencies were captured via subsequent fusion and bidirectional reconstruction. Specifically, each Transformer encoder's output is used to reconstruct the alternate omic through a shared latent alignment space (Fig 1D), with cycle-consistency loss encouraging mutual consistency across both reconstruction paths. This loss is computed as the L2 norm between the input omics and their reconstructions from the opposing branch, ensuring that learned features are informative and transferable across modalities. Compared to traditional L1/L2 reconstruction alone, this cycle-consistency approach enforces deeper structural alignment, especially under unpaired sample scenarios.

The outputs of the two Transformer encoders were concatenated and passed through a linear fusion layer to generate a unified latent representation. This vector served two branches (Fig 1C): one entered a multilayer perceptron (MLP) classifier for AD prediction; the other was passed into two modality-specific Transformer decoders, which received the shared latent vector and reconstructed omics data via corresponding autoencoder decoders. Each decoder mirrored the architecture of its encoder counterpart and incorporated cross-attention to the fusion layer output, enabling conditional reconstruction.

To improve interpretability, AE-Trans incorporates the counterfactual integrated gradients (CIG) method (Fig 1E). Unlike standard integrated gradients, CIG contrasts predictions against a biologically meaningful reference (e.g., zero-expression baseline), attributing importance in a manner that emphasizes causal relevance. This enables robust quantification of each input feature's contribution to model predictions and facilitates the discovery of disease-relevant biomarkers. In downstream validation, we fine-tuned the pretrained AE-Trans encoder on two independent RNA-seq datasets (GSE118553 and GSE29378) to adapt it to cohort-specific distributions. The resulting RNA-based latent representations were then used to train a logistic regression classifier, achieving high AD prediction accuracy (AUC = 0.92 and 0.89, respectively). Furthermore, unsupervised clustering on the latent space revealed patient subgroups with significant survival differences (Fig 1F), suggesting that AE-Trans-derived features capture biologically meaningful variation with clinical relevance.

## Feature dimensionality reduction

Due to the high dimensionality of the data, which significantly increases the complexity of feature integration, we used an AE (autoencoder) model for dimensionality reduction. AE is an unsupervised dimensionality reduction technique that can be considered a nonlinear alternative to PCA. It compresses the data from the high-dimensional feature space into a lower-dimensional feature space, known as the latent space, through the encoder. The latent space can then be reconstructed back to the original input through the decoder [24]. AE model consists of two hidden layers, which can be represented by the following equations:

$$H_1 = f_1 \left( W_1 X + b_1 \right) \tag{1}$$

$$Z = f_1 \left( W_2 H_1 + b_2 \right) \tag{2}$$

Here, $X$ is the input to the encoder, $W_1$ and $W_2$ are the weight matrices from the input layer to the first hidden layer and from the first hidden layer to the latent space, respectively. $b_1$ and $b_2$ are the corresponding bias vectors, and $b_1$ and $b_2$ are

their activation functions. In this paper, the activation function for the encoder is ReLU for both layers. $H_1$ is the output of the first hidden layer, and $Z$ is the output of the second hidden layer, which represents the low-dimensional latent space. The encoder is primarily responsible for encoding RNA and methylation data into a low-dimensional representation, effectively compressing the input data while retaining the key features of both RNA and methylation.

Our decoder also consists of two hidden layers and can be viewed as performing the inverse operation of the encoder, reconstructing the low-dimensional latent space back into the original feature space. The formulas for the decoder are as follows:

$$H_2 = f_3 \left( W_3 Z' + b_3 \right) \tag{3}$$

$$\hat{X} = f_4 \left( W_4 H_2 + b_4 \right) \tag{4}$$

In the formula, $W_3$ and $W_4$ are the weight matrices from the latent space to the first hidden layer and from the first hidden layer to the original feature space, respectively. $b_3$ and $b_4$ are their corresponding bias vectors, and and are the activation functions. In this study, $f_3$ is the ReLU function, and $f_4$ is the Sigmoid function. $H_2$ represents the output of the first hidden layer in the decoder, while $\hat{X}$ is the reconstructed output, representing the reconstructed features of the original input space.

## Feature fusion and attention mechanism

The Transformer model is a deep learning model that uses a self-attention mechanism to process input data represented as a set of embeddings, similar to the AE [25]. In AE-Trans, the Transformer module is used to further capture the global dependencies between RNA and DNA methylation data, serving as the core of our method. The input data first passes through a positional encoding module, which provides positional information to the Transformer. The formula for positional encoding is as follows:

$$PE_{(pos,2i)} = \sin \left( \frac{pos}{10000^{2i/d_{model}}} \right) \tag{5}$$

$$PE_{(pos,2i+1)} = \cos \left( \frac{pos}{10000^{2i/d_{model}}} \right) \tag{6}$$

Here, $pos$ represents the position, $d_{model}$ is the dimension index, and is the dimensionality of the input data. Positional encoding is added to the input features, allowing the model to capture dependencies between different features and adjust the weights based on the feature order.

After positional encoding, the data is passed into the Transformer encoder layer, which consists of a Multi-head Self-Attention mechanism, a Feed-Forward Network (FFN), Residual Connections, and Layer Normalization. First, the Multi-head Self-Attention mechanism enables the model to focus on different parts of the input data simultaneously through multiple attention heads. The formula for multi-head attention is as follows:

$$\text{Attention} \left( Q, K, V \right) = \text{softmax} \left( \frac{QK^T}{\sqrt{d_k}} \right) V \tag{7}$$

Here, Q, K, V represent the query, key, and value matrices, respectively, and $d_k$ is the dimension of the key. By assigning different attention weights to various features, this mechanism captures the dependencies between different features. The

multi-head attention mechanism can interpret the global structure of data from multiple perspectives. For RNA and methylation data, the model is thus able to establish cross-omics relationships between genetic and epigenetic features.

After passing through the self-attention layer, the data enters the Feed-Forward Network (FFN), which consists of two linear layers and a ReLU activation function. It can be represented as:

$$\text{FFN}(x) = \max(0, W_1 x + b_1) W_2 + b_2 \tag{8}$$

$W_1$ and $W_2$ are the weight matrices for the linear transformations, $b_1$ and $b_2$ are the bias terms。 In our AE-Trans model, the Transformer module is key to capturing global dependencies between RNA and DNA methylation data. First, the data undergoes positional encoding, followed by the multi-head self-attention mechanism, which assigns attention weights to different features and captures cross-omics relationships. After the self-attention layer, the data passes through a Feed-Forward Network (FFN) composed of two linear layers with ReLU activations, along with residual connections and layer normalization to enhance stability and prevent vanishing gradients. Our Transformer encoder consists of three layers, progressively extracting higher-order associations from the data. The parallel encoders process fused RNA and methylation features, which are then combined for further integration.

The output is passed to a Transformer decoder, which, like the encoder, includes self-attention, FFN, residual connections, and normalization. The decoder helps reconstruct the input, improving the model's understanding of the data. Additionally, an autoencoder decoder reconstructs the RNA and methylation data, preserving the input information while performing classification. Finally, the output from the Transformer is reduced through fully connected layers, activated by ReLU, and passed to a final Sigmoid-activated layer to predict the probability of Alzheimer's disease. This structure efficiently integrates high-dimensional data, leveraging the Transformer's ability to capture global dependencies for accurate classification.

### Local explanation of the AE-Trans model based on counterfactual integrated gradients attribution

To understand the contribution of each input feature in the model's prediction process, we applied the Integrated Gradients method for local interpretation of our model [26]. Unlike traditional gradient methods, Integrated Gradients accumulate the gradient changes of the input features from a baseline input $x'$ to the actual input $x$ through integration, providing a more comprehensive capture of the feature contributions to the model's output.

First, a baseline input $x'$ is chosen, typically an all-zero or all-one vector, ensuring that the model's output is independent of the feature inputs. Then, the contribution of the change from the baseline $x'$ to the actual input $x$ to the model's output y is calculated. The formula for Integrated Gradients is as follows:

$$\text{IntegratedGrad}_i(x) = (x_i - x'_i) \int_0^1 \frac{\partial F(g(k))}{\partial x_i} dk \tag{9}$$

Here, $F(x)$ represents the model's prediction function, $x_i$ is the iii-th feature of the input, and g($k$) is a path from the baseline $x'$ to the input $x$. Typically, the path g($k$) is set as a straight line, which is defined as:

$$g(k) = (1 - k)x' + kx \tag{10}$$

Along this path, the Integrated Gradients can be expressed as:

$$\text{IntegratedGrad}_i(x) = (x_i - x'_i) \int_0^1 \frac{\partial F((1\text{-}k) x' + kx)}{\partial x_i} dk \tag{11}$$

Furthermore, we approximate the integral by averaging the gradients at multiple discrete points along the path. Thus, the formula for Integrated Gradients can be expanded into a summation form as follows:

$$\text{IntegratedGrad}_i(x) = (x_i - x'_i) \frac{1}{n} \sum_{j=1}^{n} \partial F\left(\left(1 - \frac{j}{n}\right) x' + \frac{j}{n} x\right) / \partial x_i \tag{12}$$

By calculating the average gradient at each point from the baseline input to the actual input using Integrated Gradients, this method provides a more accurate estimate of feature contributions, rather than relying on a single-point gradient. In our model, the input consists of RNA and methylation data, and we use Integrated Gradients to compute the contribution of each gene expression level and methylation site to the model's prediction. This captures how each feature influences the prediction process from the baseline to the actual input, revealing the importance of RNA and methylation features in the classification task.

After calculating the Integrated Gradients for all samples, we compute a weighted average of each feature's gradients to derive the feature importance rankings. These rankings are then used for further biological analysis to identify which features play a key role in Alzheimer's disease prediction. By using Integrated Gradients, we not only provide an explanation for our model but also reveal the contribution of each feature along the path from baseline to actual input, avoiding the limitations of single-point gradients. This offers a more robust interpretability framework, helping us better understand the decision-making process of the model in Alzheimer's disease prediction.

## Evaluation metrics

In this study, we evaluated the model using five-fold cross-validation and a test set. Accuracy and Area Under the ROC Curve (AUC) were used as evaluation metrics.

Accuracy measures the proportion of correct predictions made by the model and is calculated using the following formula:

$$\text{Accuracy} = \frac{TP + TN}{TP + TN + FP + FN} \tag{13}$$

The AUC evaluates the overall performance of the model across different classification thresholds, reflecting the model's ability to distinguish between positive and negative samples. The AUC is calculated based on the ROC curve, which is plotted by computing the False Positive Rate (FPR) and True Positive Rate (TPR) at various threshold levels. The formula for AUC can be expressed as the integral of the ROC curve:

$$\text{TPR} = \frac{TP}{TP + FN} \tag{14}$$

$$\text{FPR} = \frac{FP}{FP + TN} \tag{15}$$

$$\text{AUC} = \sum_{i=1}^{n-1} (FPR_{i+1} - FPR_i) \frac{TPR_{i+1} + TPR_i}{2} \tag{16}$$

These values are used to compute metrics like the True Positive Rate (TPR) and False Positive Rate (FPR), which are then used to construct the ROC curve and calculate the AUC, providing a comprehensive evaluation of the model's classification performance.

### Training and testing

During the training and evaluation process, the model uses five-fold cross-validation to enhance its generalization ability. For each fold, the training and validation sets are split based on the labels, ensuring balanced positive and negative samples. The model is trained on the training set, while performance is evaluated on the validation set to adjust model parameters. The loss function consists of a classification loss, two reconstruction losses, and two cycle-consistency reconstruction losses: binary cross-entropy (BCE) for classification loss and mean squared error (MSE) for reconstruction losses.

During each fold's training process, we use the Adam optimizer to update model parameters, with an initial learning rate of 0.001, and apply L2 regularization to prevent overfitting. Additionally, a ReduceLROnPlateau learning rate scheduler is used, which dynamically adjusts the learning rate by monitoring changes in validation loss, ensuring fast convergence and avoiding local optima. After training, the model's performance is evaluated on the validation set using Accuracy and AUC.

In the final evaluation stage, the model was applied to an independent internal test set as well as an external test set to assess its generalization performance on previously unseen data. The predictions are further analyzed using Accuracy and AUC, and the model's predictive performance is visualized through a confusion matrix.

## Results

### Performance comparison of AE-Trans with baseline methods

To evaluate the performance of AE-Trans, we benchmarked it against six established models: three classical machine learning algorithms—Random Forest (RF), Naive Bayes (NB), and Logistic Regression (LR)—and three deep learning methods including DEG-DMP-DNN [12], AE-XGBoost [7], and Deep Belief Network (DBN) [13]. All models were trained and evaluated on harmonized RNA-seq and DNA methylation data from GSE33000, GSE44770, and GSE80970. Data were normalized and aligned to a common 14,926-gene feature space as described in the preprocessing section.

Model hyperparameters were tuned using five-fold cross-validation on the training split of GSE33000, and then fixed for both internal and external evaluation. For cross-validation, AE-Trans achieved an average Accuracy of 0.9571 and AUC of 0.9883 (Fig 2, Table 1), outperforming AE-XGBoost by 3.9% in Accuracy and LR by 3.5% in AUC. On an internal held-out test set (20% of GSE33000 + GSE44770 + GSE80970), AE-Trans reached 0.9736 Accuracy and 0.9910 AUC, exceeding AE-XGBoost by 7.3% in Accuracy and 4.7% in AUC. The associated confusion matrix (Fig 2D) confirmed robust classification of both AD and control samples.

To assess generalizability, we applied the pretrained AE-Trans model to an external test set comprising middle temporal gyrus RNA-seq data from GSE132903 and methylation data from a distinct cortical region (GSE80970 subset). No further training was performed. First, we tested AE-Trans on a cross-region unpaired dataset, which included RNA-seq data from GSE132903 and methylation data from the superior temporal gyrus in GSE80970, representing data from different brain regions. The model maintained strong performance (External test1; Accuracy = 0.7389, AUC = 0.8432), outperforming DBN by 2.2% in Accuracy and LR by 5.5% in AUC (Table 1; Fig 2A–B). Notably, all features and thresholds used for internal evaluation were retained during external testing, ensuring comparability. It is worth noting that external dataset is derived from different brain regions, which may increase the risk of model overfitting. Therefore, when making predictions across brain regions, we need to pay special attention to adjusting the model parameters.Next, we validated the model on an unpaired same-region dataset from the frontal cortex, comprising 181 transcriptomes (GSE5281, GSE15222, GSE12685, GSE36980, GSE48350) and 147 methylomes (GSE59685, GSE66351). AE-Trans demonstrated strong generalization with an accuracy of 0.8732 and an AUC of 0.9396, confirming its ability to generalize within the same brain region (External test2; Fig 2E; Table 2).Finally, to assess the model's ability to handle paired data, we tested it on a real-world paired dataset from the frontal cortex (GSE110732), which contained 25 matched transcriptome and methylation samples. AE-Trans achieved excellent results (Accuracy = 0.8800, AUC = 0.9267), showcasing its effectiveness in transferring knowledge from unpaired datasets to paired data (Paried dataset; Fig 2E; Table 2). In contrast to the cross-brain region

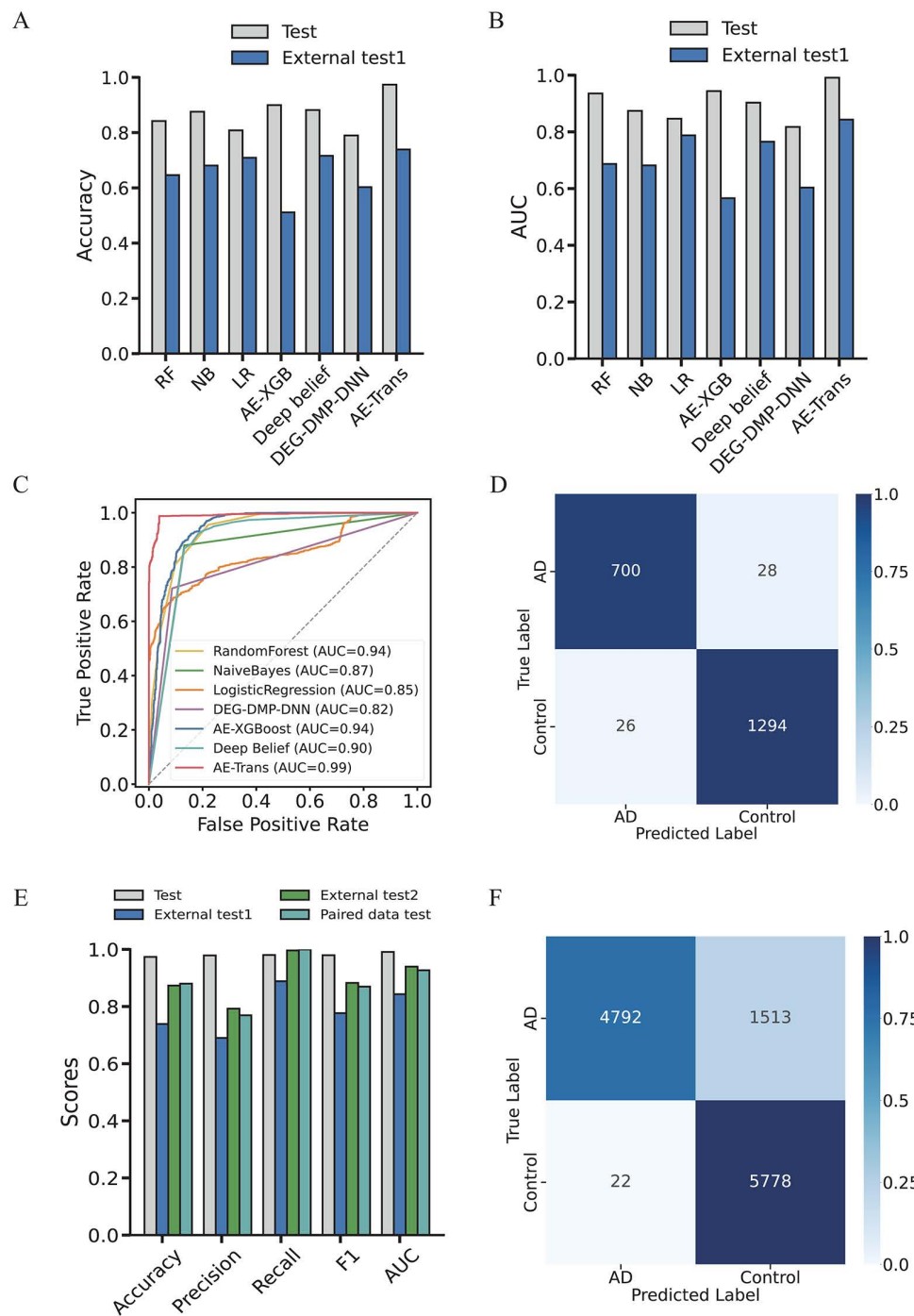

**Fig 2. Performance and confusion matrices of AE-Trans and comparative algorithms.** Fig 2. Comprehensive performance evaluation and generalizability analysis of AE-Trans. The model was benchmarked against multiple baseline methods across internal and external cohorts. (A) Comparison of Accuracy. Bar charts representing the classification accuracy of AE-Trans versus six comparative algorithms, demonstrating consistent superiority. (B) Comparison of AUC scores. Quantitative comparison of Area Under the Curve (AUC) values, highlighting the robust discriminative power of the proposed framework. (C) ROC curve analysis. Receiver Operating Characteristic (ROC) curves illustrating the diagnostic performance across different validation sets. (D) Cross-region external validation. Confusion matrix showing the classification results on External test 1, which comprises unpaired samples from geographically distinct regions. (E) Multi-scenario performance metrics. Summary of key performance indicators across three external validation scenarios, including cross-region, same-region, and paired datasets. (F) Same-region external validation. Confusion matrix displaying the model's predictive accuracy on External test 2, consisting of unpaired samples from the same geographic region.

**Table 1. Performance Comparison of AE-Trans with Other Methods.**

| Methods | Parameters | Split | Accuracy | Precision | Recall | F1 | AUC |
|---|---|---|---|---|---|---|---|
| AE-XGBoost | 15,371,584 | CV | 0.9178 | 0.9362 | 0.9134 | 0.9246 | 0.9382 |
| | | Test | 0.8994 | 0.9215 | 0.8871 | 0.9040 | 0.9436 |
| | | External | 0.5113 | 0.5731 | 0.4822 | 0.5234 | 0.5660 |
| DEG-DMP-DNN | 31,214,337 | CV | 0.7924 | 0.8461 | 0.7695 | 0.8061 | 0.8139 |
| | | Test | 0.7896 | 0.8320 | 0.7552 | 0.7918 | 0.8173 |
| | | External | 0.6026 | 0.6284 | 0.5782 | 0.6021 | 0.6032 |
| RF | 768,898 | CV | 0.8743 | 0.9428 | 0.8592 | 0.8941 | 0.9424 |
| | | Test | 0.8418 | 0.8418 | 0.9376 | 0.8083 | 0.9352 |
| | | External | 0.6462 | 0.6336 | 0.7304 | 0.6786 | 0.6869 |
| NB | 119,410 | CV | 0.8486 | 0.8997 | 0.8611 | 0.8794 | 0.8499 |
| | | Test | 0.8755 | 0.9236 | 0.8795 | 0.9010 | 0.8739 |
| | | External | 0.6808 | 0.6461 | 0.8306 | 0.7269 | 0.6817 |
| LR | 29,853 | CV | 0.8166 | 0.9632 | 0.7439 | 0.8374 | 0.9533 |
| | | Test | 0.8080 | 0.8035 | 0.9455 | 0.8067 | 0.8463 |
| | | External | 0.7089 | 0.6622 | 0.4799 | 0.4226 | 0.7877 |
| Deep Belief | 10,097,107 | CV | 0.9002 | 0.9186 | 0.8923 | 0.9052 | 0.9138 |
| | | Test | 0.8818 | 0.9062 | 0.8724 | 0.8889 | 0.9028 |
| | | External | 0.7161 | 0.6542 | 0.7915 | 0.7212 | 0.7652 |
| AE-Trans | 377,737,973 | CV | 0.9571 | 0.9749 | 0.9582 | 0.9664 | 0.9883 |
| | | Test | **0.9736** | **0.9788** | **0.9803** | **0.9796** | **0.9910** |
| | | External | **0.7389** | **0.6901** | **0.8884** | **0.7768** | **0.8432** |

**Table 2. Model Performance on Cross-Region, Same-Region, and Paired External Test Datasets.**

| DataSet | Accuracy | Precision | Recall | F1 | AUC |
|---|---|---|---|---|---|
| Test | 0.9736 | 0.9788 | 0.9803 | 0.9796 | 0.9910 |
| External test1 | 0.7389 | 0.6901 | 0.8884 | 0.7768 | 0.8432 |
| External test2 | 0.8732 | 0.7925 | 0.9962 | 0.8827 | 0.9396 |
| Paired data test | 0.8800 | 0.7692 | 1.0 | 0.8696 | 0.9267 |

dataset External test1, the within-brain region external validation set and the genuine paired data demonstrate a relatively lower risk of overfitting.

Together, these results demonstrate that AE-Trans not only excels in classification tasks under matched conditions but also generalizes reliably across anatomical regions and datasets—validating its utility in real-world multi-omics applications for Alzheimer's disease. When dealing with such studies, we should pay attention to the differences in data sets and adjust parameters to reduce the risk of overfitting.

## Interpretability of AE-Trans reveals biologically relevant features for AD prediction

To investigate the biological underpinnings of AE-Trans predictions, we applied Integrated Gradients (IG) to attribute model outputs to individual RNA and DNA methylation features. Attribution scores were computed per sample and averaged over five-fold cross-validation to ensure stability. As shown in Fig 3A and 3B, IG scores displayed a highly skewed

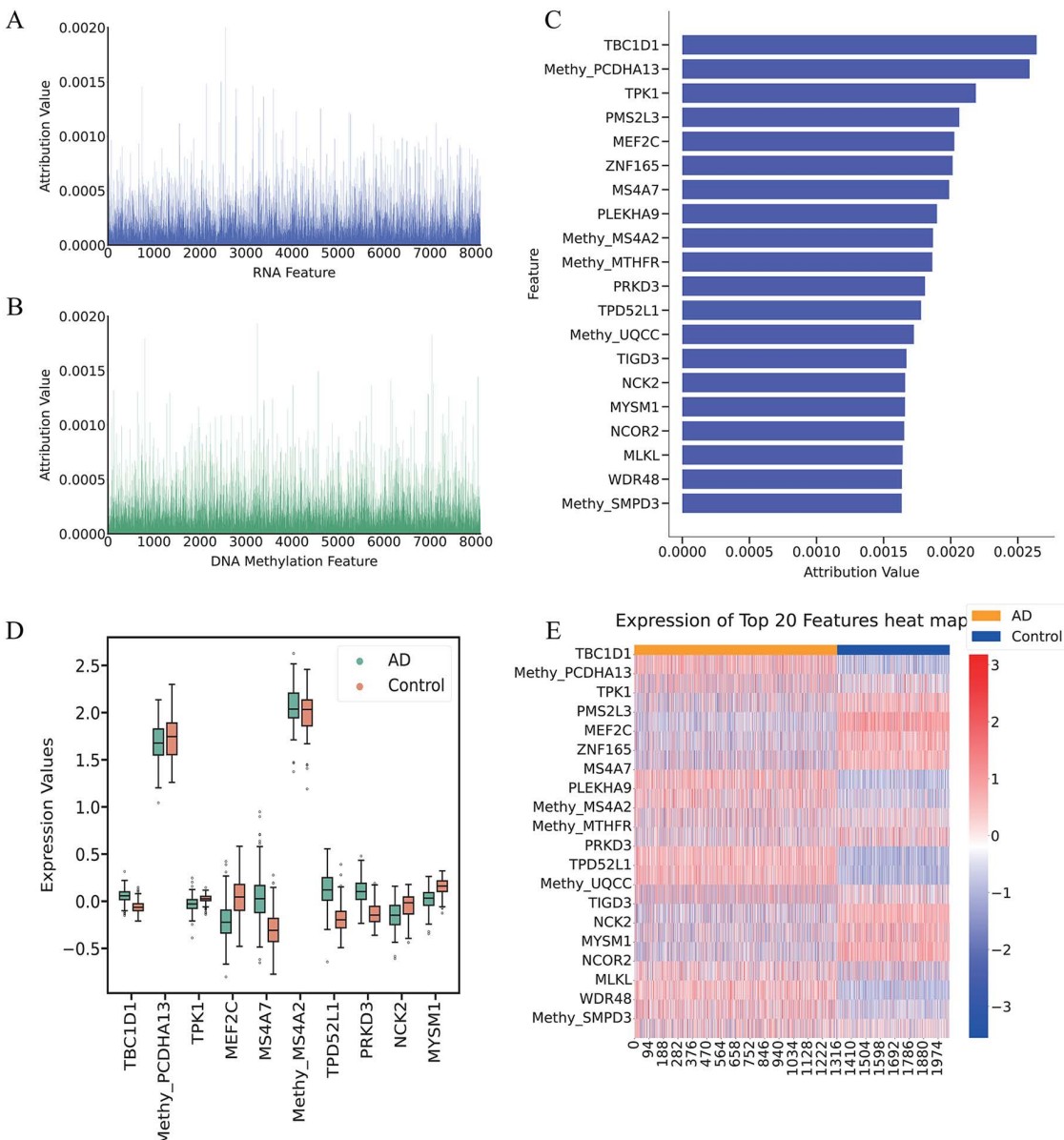

**Fig 3. Interpretable AE-Trans Analysis via Counterfactual Integrated Gradients with Key Feature Highlight.** The model utilizes Integrated Gradients (IG) to attribute predictive outputs to specific multi-omics features. (A-B) Global attribution profiles. Density plots showing the highly skewed distribution of IG scores for RNA (A) and DNA methylation (B) features, where the majority of predictive weight is concentrated in a small subset of features. (C) Quantitative ranking of top features. Bar charts illustrating the top 20 biomarkers prioritized by their average attribution scores and ranking stability across five-fold cross-validation. (D) Statistical significance of top biomarkers. Comparison of expression and methylation levels between AD and Control groups for the top-ranked features, with significance determined by Wilcoxon rank-sum tests and Benjamini-Hochberg FDR correction. (E) Heatmap of candidate biomarkers. Visualization of differential expression and methylation patterns for the identified top features, showing clear group separation and consistent biological signals across AD and non-AD samples.

distribution: over 80% of cumulative attribution was concentrated in fewer than 10% of features, suggesting a long-tailed relevance pattern.

From this distribution, we selected the top 20 features based on their average attribution scores and consistent ranking within the top 5% across folds (Fig 3C). This number was chosen based on a clear inflection point observed in the attribution curve and served as a balance between interpretability and biological tractability.

To validate their biological relevance, we compared the expression or methylation levels of these top features between AD and control samples in the test set. Statistical significance was assessed using the Wilcoxon rank-sum test with Benjamini-Hochberg FDR correction; features with adjusted p-values < 0.05 were deemed significant (Fig 3D). Expression/ methylation heatmaps showed clear group separation (Fig 3E).

We further examined nine representative features for biological interpretation, selected based on high IG attribution and reported relevance to AD or neurobiological processes [27–32]. Importantly, we checked that the direction of attribution aligned with observed trends. For instance, MEF2C was downregulated in AD and assigned a negative IG score, consistent with its known neuroprotective role [30]. Conversely, MS4A7 and TBC1D1 were upregulated in AD and assigned strong positive attributions, consistent with prior links to inflammation and metabolism [27,28,31,32].

Functionally, the identified features encompass key AD-associated mechanisms: TBC1D1 and PRKD3 are linked to glucose metabolism and tau-related signaling [27–30];; MS4A7 and MS4A2 are members of immune-regulatory families [30–32]; MEF2C and NCK2 are involved in synaptic regulation and have been identified in AD GWAS [30]; MYSM1 participates in DNA repair, and PCDHA13 is implicated in depression-related epigenetic changes relevant to AD comorbidity.

Finally, Gene Ontology enrichment on the top 100 ranked features, performed via Metascape using the 14,926 shared genes as background, revealed overrepresentation of immune signaling, hormone response, and neurodevelopmental pathways. Taken together, these results confirm that AE-Trans identifies features with both predictive value and biological coherence. While attribution does not imply causality, the convergence of statistical significance, mechanistic alignment, and literature support strengthens the interpretability and translational relevance of our multi-omics model.

## Mechanistic insight and robustness validation of interpretable features identified by AE-Trans

To evaluate the diagnostic utility of features prioritized by AE-Trans, we constructed a logistic regression classifier using the top 200 features ranked by average Integrated Gradients (IG) attribution scores. This threshold was determined by a systematic cumulative attribution analysis, where the top 200 features consistently accounted for over 90% of total model attribution weight. Quantitative evaluation of alternative thresholds (e.g., 100, 500) indicated that the 200-feature cutoff optimized the balance between predictive performance and model parsimony.

We compared the AE-Trans-based classifier with four classical feature selection approaches: random selection, F-score, coefficient of variation (CV²), and principal component analysis (PCA), using logistic regression as a unified backbone. AE-Trans achieved the highest AUC (0.9749), outperforming the next-best method (F-score) by 3.1% (Fig 4A–B). The observed performance gains are attributed to AE-Trans's unique capacity to capture complex nonlinear dependencies and integrate cross-modality patterns effectively through its Transformer architecture—capabilities inherently limited in conventional linear or univariate selection strategies.

We also benchmarked against conventional statistical biomarkers. Differentially expressed genes (DEGs) and methylated sites (DMSs) were identified using fold-change and FDR-adjusted p-value thresholds from prior studies [12], and used to build a logistic regression classifier (Fig 4C–E). The resulting AUC (0.9359) was 3.9% lower than AE-Trans. This suggests that while DEGs and DMSs capture statistically significant group-level differences, AE-Trans identifies discriminative feature combinations that optimize classification boundaries—even if some features are not individually significant. To evaluate feature robustness, we performed stepwise ablation experiments by removing top-ranked features from the input and retraining the AE-Trans model. Elimination of the top 20 features led to a 4–6% drop in accuracy; removing top 21–50 resulted in smaller declines (0.5–4%), and further removal (top 51–100) showed diminishing performance losses

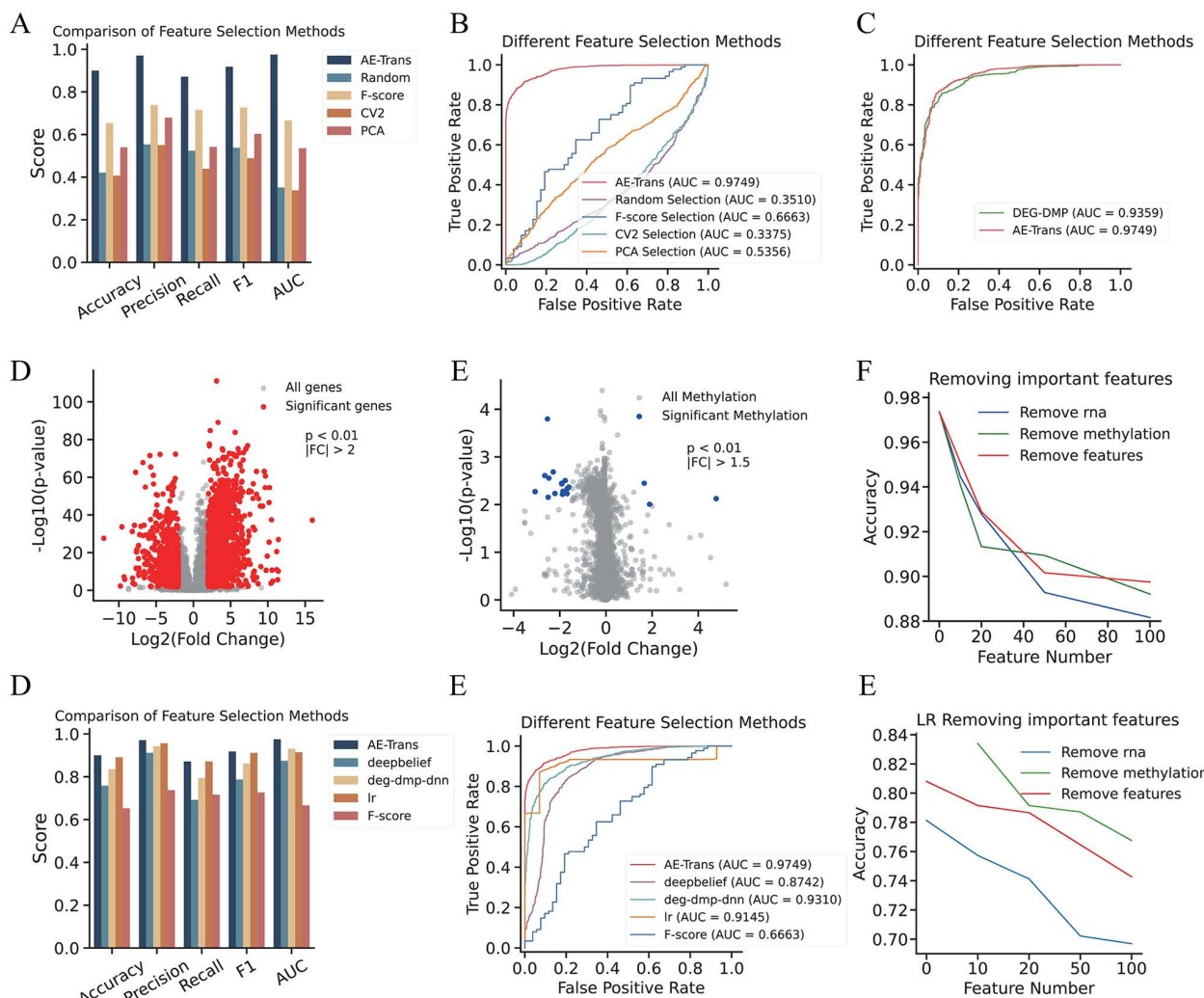

**Fig 4. Comparative Performance Analysis of Logistic Regression Using Selected Important Features Versus Alternative Feature Selection Methods.** This Fig demonstrates the superior diagnostic power of AE-Trans-prioritized features and their generalizability across different machine learning backbones. (A–B) Benchmark of feature selection methods. ROC curves (A) and quantitative comparison (B) of logistic regression (LR) classifiers built using features selected via AE-Trans (top 200), F-score, CV², PCA, and random selection. (C–E) Comparison with statistical biomarkers. Performance metrics and ROC analysis comparing AE-Trans features against traditional differentially expressed genes (DEGs) and methylated sites (DMSs), highlighting the advantage of nonlinear feature integration. (F) Feature ablation study on AE-Trans. Stepwise removal of top-ranked features (Top 1–20, 21–50, 51–100) illustrates the concentration of predictive information within the highest-scoring biomarkers. (G) Generalizability of AE-Trans explainability. Performance improvement (Accuracy, Precision, Recall, F1, AUC) of baseline models—DeepBelief, DEG-DMP-DNN, and LR—after incorporating AE-Trans-selected features. (H) AUC enhancement across architectures. Comparison of AUC scores for various models before and after applying AE-Trans-based feature selection, showing marked improvements in deep learning and linear backbones. (I) Feature ablation on LR backbone. Validation of feature importance using a logistic regression model, where the removal of methylation and key omics features leads to significant performance degradation.

(~0.2–2%) (Fig 4F). These findings confirm that AE-Trans concentrates predictive information in a compact subset of features. To assess the effectiveness of AE-Trans's explainability methods, we applied these methods to other baseline models such as DeepBelief, DEG-DMP-DNN, and Logistic Regression (LR). Fig G compares the performance of AE-Trans's explainability methods on these models before and after feature selection. After applying feature selection, DeepBelief,

DEG-DMP-DNN, and LR all showed significant improvements across multiple metrics, including Accuracy, Precision, Recall, F1, and AUC. AE-Trans's feature selection particularly enhanced DeepBelief, boosting its performance across the board. Fig H further demonstrates this improvement, showing that DeepBelief (AUC = 0.8742), LR (AUC = 0.9145), DEG-DMP-DNN (AUC = 0.9310) and AE-XGBoost (AUC = 0.9382) both saw notable increases in AUC after applying AE-Trans's explainability methods, whereas other models like F-score did not exhibit as marked an improvement, highlighting AE-Trans's particular strength in optimizing feature selection for some models. Fig I presents the results of feature ablation experiments conducted using LR. The removal of important features like methylation and MA led to a noticeable drop in accuracy, further underscoring the importance of AE-Trans's feature selection approach. These results suggest that AE-Trans's explainability methods improve the performance of other models, particularly by identifying the most relevant features for classification.

In conclusion, AE-Trans's explainability methods not only enhanced its own performance but also significantly boosted the performance of other baseline models, such as DeepBelief, DEG-DMP-DNN, and LR, making them more robust and efficient. This demonstrates the versatility and effectiveness of AE-Trans in enhancing model interpretability and feature selection across different machine learning methods.

We next investigated the biological coherence of the selected features. GO and KEGG enrichment analyses performed on the top 100 ranked features revealed significant overrepresentation of pathways central to AD pathogenesis, including immune activation (e.g., T cell receptor signaling), metabolic signaling (e.g., glucose metabolism), and synaptic regulation (e.g., axon guidance) (Fig 5A–B). These enriched terms underscore the relevance of AE-Trans-derived features to known pathophysiological processes in Alzheimer's disease. Co-expression analysis was then performed among the top 20 RNA features. The Pearson correlation threshold (r > 0.75) for co-expression analysis was selected based on established criteria for robust network construction, and the functional co-involvement of identified modules was further corroborated by extensive literature evidence. Two distinct modules emerged (Fig 5C). The first included NCK2 and MEF2C, both implicated in synaptic plasticity and identified in AD genetic studies [33,34]. The second module comprised TBC1D1, MLKL, PRKD3, and MS4A7, linking glucose dysregulation [28], necroptosis [35]. [36,37], CREB-mediated memory pathways [38,39], and neuroinflammatory activation [40]. Network topology analysis (Fig 5D) highlighted genes such as NCK2 and TBC1D1 as central hubs, suggesting their potential roles as integrative regulators of disease-relevant molecular programs.

## Generalization of AE-Trans representations to single-modality RNA and prognostic subtyping

To evaluate the applicability of AE-Trans in clinical scenarios where multi-omics data are limited, we applied the pretrained model to two independent RNA-seq datasets (GSE118553 and GSE29378), fine-tuning only the Transformer encoder on single-modality input.

On GSE118553, a classifier trained on AE-Trans latent representations achieved strong discriminative power (AUC = 0.92), significantly outperforming the baseline trained on raw RNA (AUC = 0.60; Fig 6A). UMAP projection of the latent space revealed a clear separation between AD and control samples (Fig 6B), demonstrating the model's ability to encode diagnostic structure.However, this separation was not absolute. On closer inspection, we observed a subset of AD patients interspersed within the control cluster—an unexpected finding that raised the possibility of hidden biological heterogeneity within the AD group.

To explore this, we focused on AD patients and applied unsupervised clustering to their latent vectors. This revealed two robust expression subtypes (Fig 6C), raising the question of whether these molecular differences might be clinically relevant. When stratifying patients by these subgroups and performing Kaplan–Meier survival analysis, we found that the two clusters exhibited significantly different survival outcomes (Fig 6D). This suggests that AE-Trans can uncover prognostically meaningful patterns even from RNA-only data.

We validated this discovery on an independent dataset, GSE29378. AE-Trans again outperformed the baseline classifier (AUC = 0.89 vs. 0.68; Fig 6E), and UMAP embedding again displayed a strong AD vs. control separation (Fig 6F), with

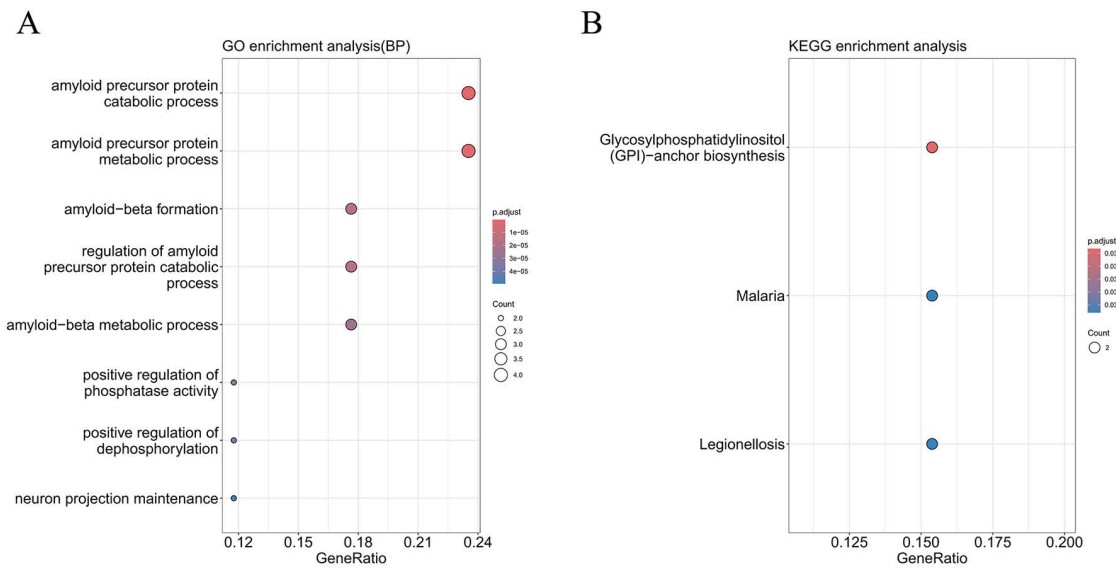

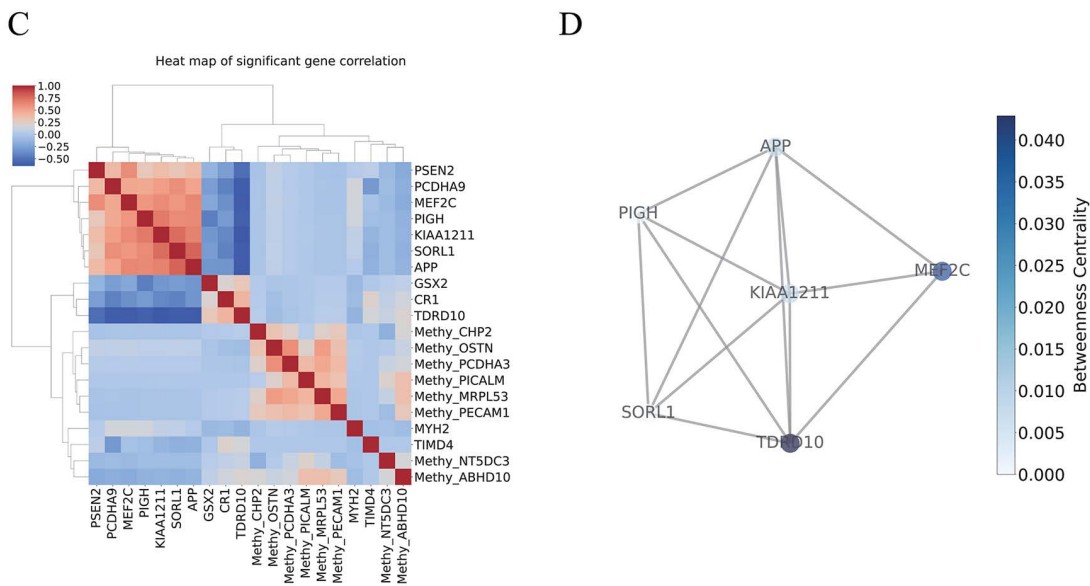

**Fig 5. Functional Enrichment and Co-expression Network Analysis Based on Important Features.** This Fig characterizes the pathophysiological relevance and systemic interactions of the top-ranked multi-omics features. (A–B) Pathway enrichment analysis. Results from Gene Ontology (GO) and KEGG enrichment analysis on the top 100 features, highlighting key AD-related processes including immune activation (T cell receptor signaling), metabolic dysregulation (glucose metabolism), and synaptic regulation (axon guidance). (C) Co-expression module analysis. Pearson correlation network (r > 0.75) of the top 20 RNA features revealing two distinct functional modules: Module 1 (e.g., NCK2, MEF2C) associated with synaptic plasticity, and Module 2 (e.g., TBC1D1, MLKL, MS4A7) linking metabolism, necroptosis, and neuroinflammation. (D) Network topology and hub gene identification. Visualization of the molecular interaction landscape where genes such as NCK2 and TBC1D1 emerge as central hubs, suggesting their roles as master regulators in the coordination of AD-relevant molecular programs.

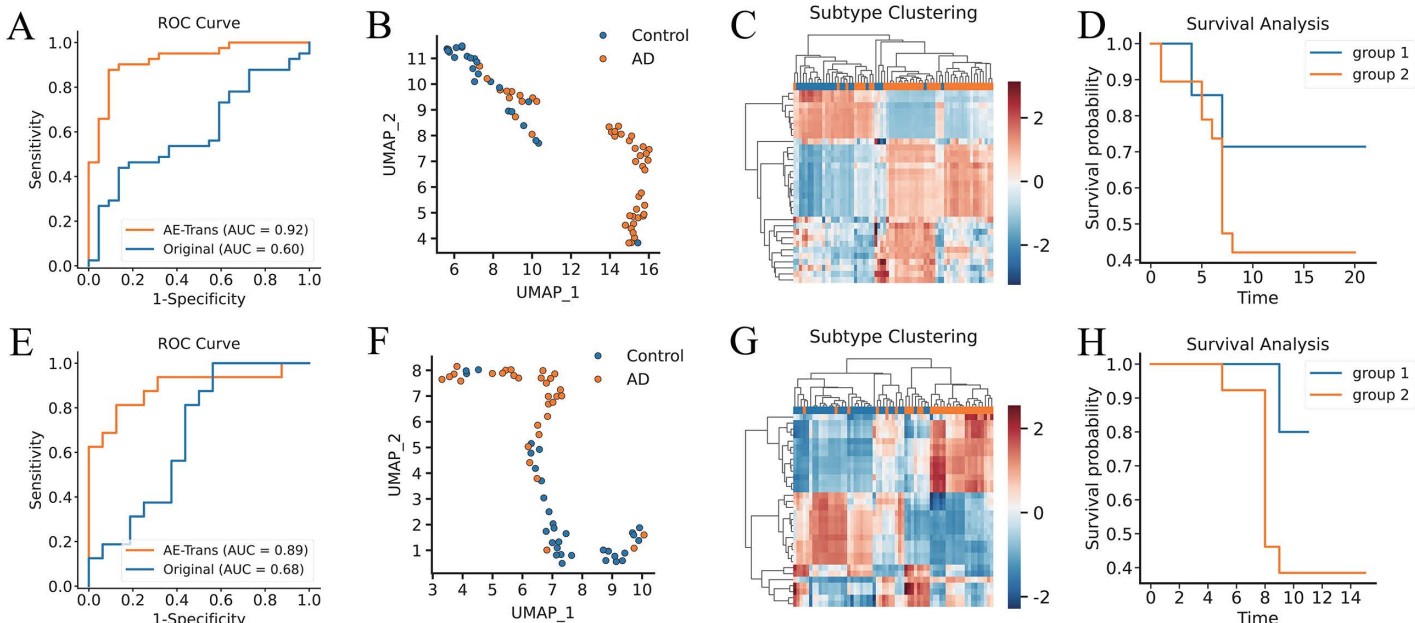

**Fig 6. AE-Trans Achieves High-Accuracy Classification and Survival Risk Stratification in RNA-Only Alzheimer's Disease Data.** AE-Trans was evaluated for its diagnostic generalizability and its ability to uncover clinically relevant biological heterogeneity using GSE118553 and GSE29378. (A, E) Diagnostic performance comparison. ROC curves for GSE118553 (A) and GSE29378 (E) show that classifiers fine-tuned on AE-Trans latent representations significantly outperform those trained on raw RNA data. (B, F) Latent space visualization. UMAP projections of the latent embeddings for GSE118553 (B) and GSE29378 (F), illustrating a clear separation between AD and control samples, while identifying a subset of AD cases with control-like molecular profiles. (C, G) Discovery of AD molecular subtypes. Unsupervised clustering of latent vectors within AD patient cohorts identifies two distinct molecular subtypes in both GSE118553 (C) and GSE29378 (G). (D, H) Clinical prognostic relevance. Kaplan–Meier survival analysis demonstrates significantly different clinical outcomes between the identified AD subtypes, confirming that AE-Trans captures prognostically meaningful biological heterogeneity even from single-modality inputs.

a few AD cases clustering near the control group. Unsupervised clustering on AD samples revealed two subtypes (Fig 6G), and survival analysis confirmed a consistent prognostic divergence (Fig 6H).

These results indicate that AE-Trans not only generalizes well to single-modality data, but also enables discovery of clinically relevant substructure—revealing potential prognostic heterogeneity within AD populations.

## Model and multi-dimensional data ablation study

We conducted ablation experiments on the omics features and different modules of the AE-Trans model. First, we validated the enhancement in prediction performance brought about by multi-dimensional data fusion. We compared the performance of the AE-Trans model using omics data fusion with that using only singgle omic data for predicting Alzheimer's disease (AD) (Fig 7). Figs 7A and 7C present a comparison of Accuracy, Precision, Recall, F1-measure, and AUC in five-fold cross-validation. The AE-Trans achieved a training accuracy of 0.957 and an AUC of 0.9883. Figs 7B and 7D show the results on the test set, where AE-Trans reported an ACC and AUC of 0.9736and 0.9910, respectively. The metrics for the omics fusion were consistently optimal, with the performance of the model using only RNA being slightly lower than that of the multi-dimensional data, showing an ACC and AUC of 0.9357and 0.9622, respectively. In contrast, using only methylation data resulted in significantly poorer outcomes, with an accuracy of 0.414 and an AUC of 0.529. Table 3 provides detailed metrics for different omics across evaluation indicators, showing that dual-channel fusion achieved approximately 3.8% and 2.8% higher Accuracy and AUC on the test set compared to RNA alone. These results

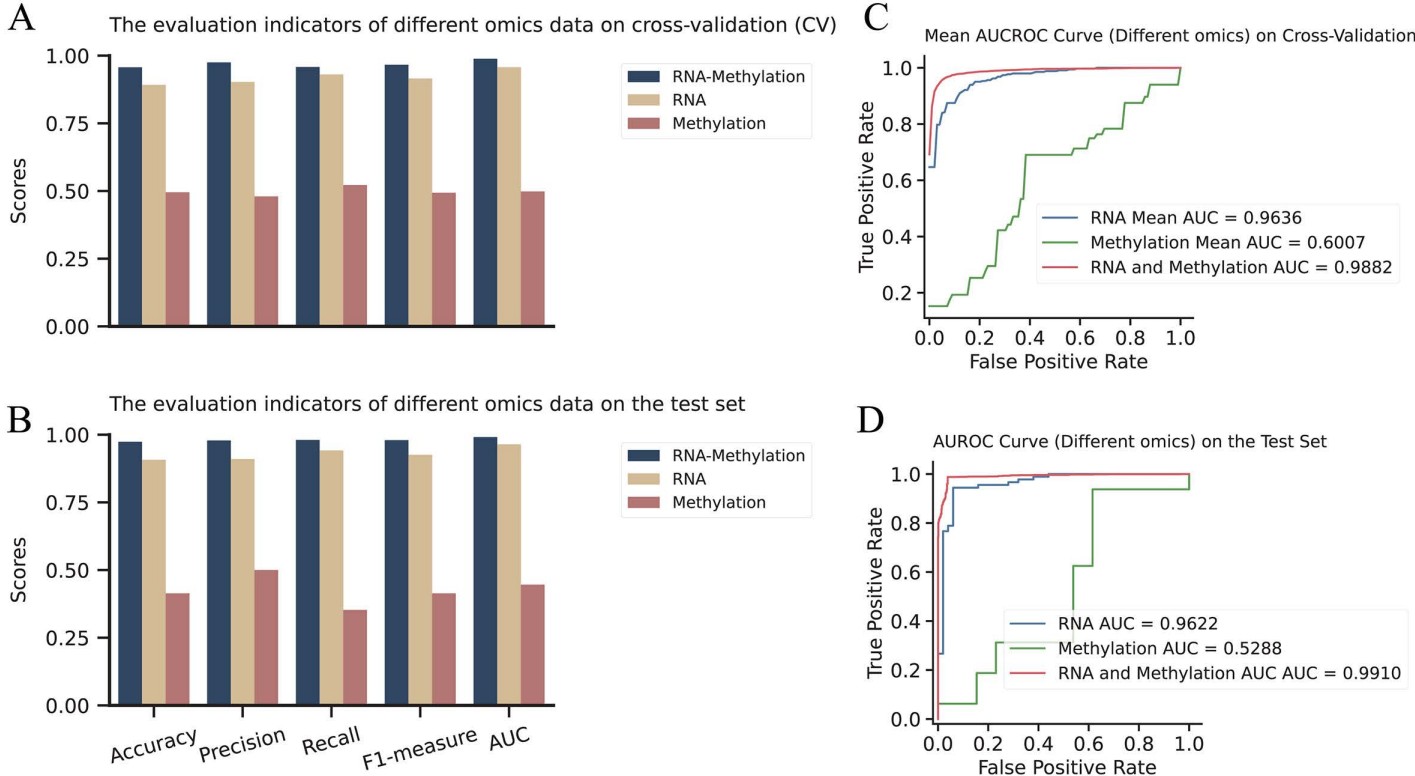

**Fig 7. Comparison of prediction performance between multi-dimensional fusion and single dimensional data.** This Fig quantifies the predictive gains achieved by integrating RNA-seq and DNA methylation data compared to using individual omics channels. (A, C) Performance benchmarking in five-fold cross-validation. Comparison of Accuracy, Precision, Recall, F1-measure, and AUC across different input configurations, showing that the dual-channel AE-Trans model (AUC = 0.9883) consistently outperforms single-modality approaches during training. (B, D) Validation on the independent test set. Results demonstrate the superior generalizability of the multi-omics fusion strategy (Accuracy = 0.9736; AUC = 0.9910) over RNA-only (Accuracy = 0.9357) and methylation-only models.

**Table 3. Performance evaluation table for different omics data.**

| Omics Data | Split | Accuracy | Precision | Recall | F1-measure | AUC |
|---|---|---|---|---|---|---|
| RNA+Methylation | CV | 0.9571 | 0.9749 | 0.9582 | 0.9664 | 0.9883 |
| | Test | 0.9736 | 0.9788 | 0.9803 | 0.9796 | 0.9910 |
| RNA | CV | 0.9100 | 0.9345 | 0.9192 | 0.9267 | 0.9637 |
| | Test | 0.9357 | 0.9655 | 0.9333 | 0.9492 | 0.9622 |
| Methylation | CV | 0.5577 | 0.5802 | 0.6014 | 0.5580 | 0.6007 |
| | Test | 0.4138 | 0.4615 | 0.3750 | 0.4138 | 0.5288 |

clearly demonstrate that multi-dimensional data fusion significantly outperforms single omic data in predicting Alzheimer's disease.

Next, we performed ablation studies on our model structure, including the removal of the Autoencoder (AE), using only a single layer of Transformer encoder and decoder layers, and excluding the masking mechanism from the AE-Trans model. The results are summarized in Table 4.

**Table 4. Model ablation performance evaluation table.**

| Model Ablation | Split | Accuracy | Precision | Recall | F1-measure | AUC |
|---|---|---|---|---|---|---|
| AE-Trans | CV | 0.9571 | 0.9749 | 0.9582 | 0.9664 | 0.988 |
| | Test | **0.9736** | **0.9788** | **0.9803** | **0.9796** | **0.9910** |
| without AE | CV | 0.9373 | 0.9403 | 0.9644 | 0.9521 | 0.9828 |
| | Test | 0.9170 | 0.9450 | 0.9250 | 0.9349 | 0.9581 |
| Only 1 Transformer layer | CV | 0.8962 | 0.9363 | 0.9044 | 0.9160 | 0.9740 |
| | Test | 0.8989 | 0.9448 | 0.8955 | 0.9195 | 0.9335 |
| without Mask | CV | 0.7652 | 0.7659 | 0.9885 | 0.8539 | 0.8456 |
| | Test | 0.9004 | 0.9415 | 0.9015 | 0.9211 | 0.8979 |

It can be observed that removing the AE resulted in slight reductions in all evaluation metrics, both in cross-validation (CV) and test sets. In contrast, using only one layer of Transformer encoder and decoder layers led to a significant decline in performance across all metrics, particularly in Accuracy and Recall. When the masking mechanism was omitted, all metrics except for Recall showed substantial decreases in CV, while Recall and AUC also dropped notably in the test set. Finally, when only the AE was retained, all metrics, except for Recall in the CV validation, were significantly lower than those of the AE-Trans method. These results indicate that each module designed in our model contributes to enhancing prediction accuracy when using multi-dimensional data fusion for predicting Alzheimer's disease.

## Discussion and conclusion

Alzheimer's disease (AD) is a complex neurodegenerative disorder that remains difficult to diagnose, particularly at early stages. In this study, we developed AE-Trans, a dual-channel Transformer-based framework capable of integrating unpaired RNA and DNA methylation data. Through multi-stage representation learning and bidirectional reconstruction, AE-Trans effectively captures complementary modality-specific signals and aligns latent features across data types. Our model consistently outperformed comparative methods across a range of classification tasks, including same-brain-region, cross-brain-region, and real paired datasets. Specifically, AE-Trans achieved an AUC of 0.9910 on the prefrontal cortex data, and reached an AUC of 0.8432 on cross-region temporal cortex data. Moreover, it attained an AUC of 0.94 on an external unpaired dataset from the same brain region, and achieved an AUC of 0.93 on a real-world paired multi-omics dataset.These results demonstrate the robustness and generalizability of our approach for AD classification. However, when conducting such studies, we must pay attention to the variations in datasets across different brain regions to minimize the risk of overfitting that might be introduced.

Beyond predictive accuracy, AE-Trans offers strong interpretability through integrated gradients, enabling attribution of classification decisions to specific transcriptomic and epigenetic features. The top-ranked genes identified through this method—such as TBC1D1, MS4A7, and MEF2C—are consistent with known AD-related pathways including neuroinflammation, metabolic stress, and synaptic regulation. Feature ablation and GO/KEGG enrichment analyses further supported the biological relevance of these markers. Notably, a logistic regression model trained solely on the top 200 attributed features achieved an AUC of 0.9749, outperforming traditional feature selection methods and differential analysis approaches, suggesting that AE-Trans identifies predictive features that extend beyond conventional statistical signals.

Importantly, we demonstrated that the representations learned by AE-Trans retain clinical utility in single-modality contexts. When applied to two independent RNA datasets, the model not only improved classification performance but also revealed latent patient subtypes with significant prognostic differences—highlighting its potential to inform clinical risk stratification and guide personalized interventions.

Nevertheless, the model operates on synthetically paired data derived from unpaired multi-omics samples. While our Cartesian pairing strategy has proven effective, it may not fully capture intra-individual dependencies present in truly paired samples. This limitation is especially relevant in single-cell or longitudinal multi-omics, where cell-specific or

temporal molecular coordination plays a critical role. Future extensions of AE-Trans could incorporate advanced strategies such as contrastive learning or domain adaptation to better align cross-modal representations under unpaired settings.

Incorporating additional data modalities (e.g., proteomics, metabolomics, imaging) and temporal dynamics from longitudinal cohorts will further improve model resolution and applicability. Although such integrations present new challenges—such as data heterogeneity and sparsity—the Transformer architecture offers an adaptable foundation for handling diverse and temporally structured inputs.

In summary, AE-Trans not only delivers high diagnostic accuracy but also provides mechanistically interpretable and clinically actionable insights. Its ability to identify risk-associated genes and stratify prognostic subtypes underscores its translational potential. These findings establish AE-Trans as a robust framework for multi-omics integration in AD research and a promising tool for advancing precision diagnostics and therapeutic development.

### Ethics approval and consent to participate

This study does not involve human participants, animal experiments, clinical trial or any other procedures requiring ethical approval.

### Code availability

All data and code are publicly available at https://github.com/bowei-color/AE-Trans.

### Supporting information

**S1 Table. Supplementary table for performance comparison of AE-Trans and other methods.**
(DOCX)

**S2 Table. Results obtained by different feature selection methods.**
(DOCX)

### Author contributions

**Conceptualization:** Kai Liao, shanshan Wu, Bowei Yan.

**Data curation:** Kai Liao, jiawei Li, Changshui chen.

**Formal analysis:** Changshui chen.

**Investigation:** Danfeng Du.

**Methodology:** Danfeng Du, jiawei Li.

**Resources:** shanshan Wu.

**Software:** jian Huang, Bowei Yan.

**Supervision:** jian Huang, Bowei Yan.

**Validation:** jian Huang, Xiaodan Fan, shanshan Wu, Bowei Yan.

**Visualization:** jiawei Li, Xiaodan Fan, Bowei Yan.

**Writing – original draft:** Xiaodan Fan, Bowei Yan, Haibo Li.

**Writing – review & editing:** Bowei Yan, Haibo Li.

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
