## [Decision Letter · Decision Letter 0]

21 Oct 2025

Interpretable Integration of Unpaired Multi-Omics for Alzheimer’s Diagnosis via Cross-Modal Transformer Reconstruction

PLOS Computational Biology

Dear Dr. Li,

Thank you for submitting your manuscript to PLOS Computational Biology. After careful consideration, we feel that it has merit but does not fully meet PLOS Computational Biology's publication criteria as it currently stands. Therefore, we invite you to submit a revised version of the manuscript that addresses the points raised during the review process.

Please submit your revised manuscript within 60 days Dec 21 2025 11:59PM. If you will need more time than this to complete your revisions, please reply to this message or contact the journal office at ploscompbiol@plos.org. Please include the following items when submitting your revised manuscript:

We look forward to receiving your revised manuscript.

Kind regards,

Samuel V. Scarpino

Academic Editor

PLOS Computational Biology

Shihua Zhang

Section Editor

PLOS Computational Biology

**Additional Editor Comment:**

I agree with the reviewers that the manuscript presents an interesting approach to explainability that could potentially be of widespread appeal in the community. However, I also agree with the reviewers that strong evidence exists that the transformer-based method presented as the backbone of the paper is overfitting. The reviewers cite two pieces of evidence in their assessment. First, we have an a priori expectation that a transformer model might overfit to the data given the vastly larger number of parameters it likely contains (as compared to other comparator methods used in the paper). Second, we can see in Table 1 that performance of the transformer-based model falls off more on the external test set than we see in the other models. That being said, it is quite interesting that the transformer-based model still seems to do the best on--as least based on accuracy--the external test set (all models struggled). I also agree with the reviewer comments that more works is needed to justify the biological relevance of the results given that the input data was not paired at the individual-level (I don't believe this is a fatal flaw, but one that should be discussed in more detail and possibly explored with sensitivity analyses using a simulation model). In addition to the reviewer comments, which I encourage the authors to pay careful attention to in their revision, I think the following additional items should be considered:

1. In Table 1, please report precision and recall in addition to accuracy (or a similar set of measures). I would also expect to see the baseline expectation reported for random guessing. It appears as though the sample sizes are the same across all models, so that column could be replaced and the sample sizes provided in a table caption. It would also be quite helpful to include an additional column listing the number of parameters in each model.

1b. I also think that Table 1 is quite hard to parse. Please consider adding shading or lines so that the reader can more quickly assess differences. You might also consider re-ordering the models such that they are presented in descending order based on external test-set performance. Be sure in the legend to explain the table fully, including how ordering was determined

2. The figure and table captions should provide more information on how to interpret the figure without the need to refer back to the text. In some cases, the titles are also too colloquial, e.g., F4 says "Accuracy..." but then many other measures of performance are provided. Similar issues appear in other figure titles. This and the above point may seem like nitpicking, but I believe it's easy to misinterpret the findings (especially when readers are skimming).

3. For Figures 4 and other generalizability results (e.g., 5 and 6), I would expect to see features included from one of the other top performing models. Presumably you could use the same feature selection procedure (showing how your approach to explainability generalizes to other complicated model structures would strengthen the paper significantly). The best solution here would be to add an additional panel with all the models in Table 1. If the authors believe that is too busy, then you could add just a single model (maybe the deep belief or LR) and then add a supplemental figure showing all the others.

I thank the authors for sending us their manuscript and will look forward to receiving a revision.

**Journal Requirements:**

4) Please ensure that all Figure files have corresponding citations and legends within the manuscript. Currently, Figures 2, and 7 in your submission file inventory do not have in-text citations. Please include the in-text citations of the figures.

5) We have noticed that you have uploaded Supporting Information files, but you have not included a list of legends. Please add a full list of legends for your Supporting Information files after the references list.

Potential Copyright Issues:

i) Figure 1A. Please confirm whether you drew the images / clip-art within the figure panels by hand. If you did not draw the images, please provide (a) a link to the source of the images or icons and their license / terms of use; or (b) written permission from the copyright holder to publish the images or icons under our CC BY 4.0 license. Alternatively, you may replace the images with open source alternatives. See these open source resources you may use to replace images / clip-art:

7) Please amend your detailed Financial Disclosure statement. This is published with the article. It must therefore be completed in full sentences and contain the exact wording you wish to be published.

8) Your current Financial Disclosure states receiving several funds. However, your funding information on the submission form doesn't indicate any funds. Please ensure that the funders and grant numbers match between the Financial Disclosure field and the Funding Information tab in your submission form. Note that the funders must be provided in the same order in both places as well.

9) Please revise your current Competing Interest statement to the standard "The authors have declared that no competing interests exist."

**Reviewers' comments:**

Reviewer's Responses to Questions

Reviewer #1: This manuscript presents AE-Trans, a dual-channel Transformer framework for integrating unpaired RNA-seq and DNA methylation data for Alzheimer's disease diagnosis. While the work addresses an important clinical need and demonstrates promising results, several methodological concerns and missing technical details limit the robustness of the findings:

- The motivation of using transformer architecture is not clear. The datasets used for the training are small for transformer model and the chance of overfitting is high.

- Does Cartesian intra-label approach truly capture inter-omics relationships as they exist in individual patients, or does it primarily learn label-specific patterns that may not generalize to real biological systems?

- It is not clear whether all baseline models (RF, NB, LR, DEG-DMP-DNN, AE-XGBoost, DBN) were trained on the same harmonized 14,926-gene feature space as AE-Trans.

- The gap in accuracy between Test (CV, Test) and external set support the idea of overfitting. This should be explained by the authors.

Reviewer #2: Your work addresses an important clinical problem and proposes an innovative technical framework that combines several promising methodologies. The work has many strengths: the interpretability framework using counterfactual integrated gradients is valuable, the identified biological pathways show reasonable concordance with AD literature, and the technical innovation of bidirectional reconstruction in Transformers has potential merit. However, there are some concerns about the experimental design and data integration strategy that must be addressed before publication:

- Creating all possible combinations between RNA-AD samples and Methylation-AD samples fundamentally violates the biological reality that each patient has unique cross-modal molecular relationships. This synthetic pairing strategy assumes all AD patients share identical RNA-methylation associations, which is biologically implausible given the known heterogeneity of Alzheimer's disease.

This approach creates several problems: 1. it generates artificial relationships that don't exist in real patients, 2. it dramatically inflates your dataset size with redundant synthetic combinations, and 3. it may explain the unusually high performance metrics that seem inconsistent with the complexity of AD diagnosis.

- Different brain regions have distinct molecular profiles and assuming AD signatures are identical across regions lacks support. This may explain the substantial performance drop in external validation.

- The results require validation in larger, independent studies before drawing clinical conclusions. The feature selection validation using the same data for both identification and testing risks overfitting.

- Several technical aspects need clarification: the "masking mechanism" mentioned in ablation studies is never properly explained, the exact implementation of your synthetic pairing during cross-validation is unclear, and batch harmonization details are insufficient for reproducibility. The sample size discrepancies between your described 80/20 split and the reported table values need resolution.

**Have the authors made all data and (if applicable) computational code underlying the findings in their manuscript fully available?**

Reviewer #1: None

Reviewer #2: Yes

PLOS authors have the option to publish the peer review history of their article (what does this mean? ). If published, this will include your full peer review and any attached files.

**Do you want your identity to be public for this peer review?** For information about this choice, including consent withdrawal, please see our Privacy Policy .

Reviewer #1: No

Reviewer #2: No

**Figure resubmission:**
---

## [Decision Letter · Decision Letter 1]

20 Feb 2026

PCOMPBIOL-D-25-01596R1

Interpretable Integration of Unpaired Multi-Omics for Alzheimer’s Diagnosis via Cross-Modal Transformer Reconstruction

PLOS Computational Biology

Dear Dr. Li,

Thank you for submitting your manuscript to PLOS Computational Biology. After careful consideration, we feel that it has merit but does not fully meet PLOS Computational Biology's publication criteria as it currently stands. Therefore, we invite you to submit a revised version of the manuscript that addresses the points raised during the review process.

We look forward to receiving your revised manuscript.

Kind regards,

Samuel V. Scarpino

Academic Editor

PLOS Computational Biology

Shihua Zhang

Section Editor

PLOS Computational Biology

**Additional Editor Comments:**

I agree with R2 that the authors should add some additional caveats regarding the risk of overfitting.

**Journal Requirements:**

1) Your manuscript is missing the following sections: Methods.  Please ensure all required sections are present and in the correct order. Make sure section heading levels are clearly indicated in the manuscript text, and limit sub-sections to 3 heading levels. An outline of the required sections can be consulted in our submission guidelines here:

2) Please amend your detailed Financial Disclosure statement. This is published with the article. It must therefore be completed in full sentences and contain the exact wording you wish to be published.

**Reviewers' comments:**

Reviewer's Responses to Questions

**Comments to the Authors:**

Reviewer #1: The authors answered my concerns carefully.

Reviewer #2: Thank you for addressing the reviewers' comments. The risk of overfitting remains a concern given the very large model size relative to the available data. Although the added external datasets and experiments are helpful, a more cautious interpretation of the results and claims would be advised.

**Have the authors made all data and (if applicable) computational code underlying the findings in their manuscript fully available?**

Reviewer #1: None

Reviewer #2: Yes

PLOS authors have the option to publish the peer review history of their article (what does this mean? ). If published, this will include your full peer review and any attached files.

**Do you want your identity to be public for this peer review?**  For information about this choice, including consent withdrawal, please see our Privacy Policy .

Reviewer #1: No

Reviewer #2: No

**Figure resubmission:**
---

## [Editor Report · Decision Letter 2]

26 Feb 2026

Dear Mr Li,

We are pleased to inform you that your manuscript 'Interpretable Integration of Unpaired Multi-Omics for Alzheimer’s Diagnosis via Cross-Modal Transformer Reconstruction' has been provisionally accepted for publication in PLOS Computational Biology.

Best regards,

Samuel V. Scarpino

Academic Editor

PLOS Computational Biology

Shihua Zhang

Section Editor

PLOS Computational Biology

---

## [Editor Report · Acceptance letter]

PCOMPBIOL-D-25-01596R2

Interpretable Integration of Unpaired Multi-Omics for Alzheimer’s Diagnosis via Cross-Modal Transformer Reconstruction

Dear Dr Li,

I am pleased to inform you that your manuscript has been formally accepted for publication in PLOS Computational Biology. Your manuscript is now with our production department and you will be notified of the publication date in due course.

With kind regards,

Zsofia Freund
